# Ischemia-Reperfusion Programming of Alzheimer’s Disease-Related Genes—A New Perspective on Brain Neurodegeneration after Cardiac Arrest

**DOI:** 10.3390/ijms25021291

**Published:** 2024-01-20

**Authors:** Ryszard Pluta, Stanisław J. Czuczwar

**Affiliations:** Department of Pathophysiology, Medical University of Lublin, 20-090 Lublin, Poland; stanislaw.czuczwar@umlub.pl

**Keywords:** cardiac arrest, cerebral ischemia, reperfusion, amyloid, tau protein, autophagy, mitophagy, apoptosis, Alzheimer’s disease, genes

## Abstract

The article presents the latest data on pathological changes after cerebral ischemia caused by cardiac arrest. The data include amyloid accumulation, tau protein modification, neurodegenerative and cognitive changes, and gene and protein changes associated with Alzheimer’s disease. We present the latest data on the dysregulation of genes related to the metabolism of the amyloid protein precursor, tau protein, autophagy, mitophagy, apoptosis, and amyloid and tau protein transport genes. We report that neuronal death after cerebral ischemia due to cardiac arrest may be dependent and independent of caspase. Moreover, neuronal death dependent on amyloid and modified tau protein has been demonstrated. Finally, the results clearly indicate that changes in the expression of the presented genes play an important role in acute and secondary brain damage and the development of post-ischemic brain neurodegeneration with the Alzheimer’s disease phenotype. The data indicate that the above genes may be a potential therapeutic target for brain therapy after ischemia due to cardiac arrest. Overall, the studies show that the genes studied represent attractive targets for the development of new therapies to minimize ischemic brain injury and neurological dysfunction. Additionally, *amyloid*-related genes expression and *tau protein* gene modification after cerebral ischemia due to cardiac arrest are useful in identifying ischemic mechanisms associated with Alzheimer’s disease. Cardiac arrest illustrates the progressive, time- and area-specific development of neuropathology in the brain with the expression of genes responsible for the processing of *amyloid protein precursor* and the occurrence of *tau protein* and symptoms of dementia such as those occurring in patients with Alzheimer’s disease. By carefully examining the common genetic processes involved in these two diseases, these data may help unravel phenomena associated with the development of Alzheimer’s disease and neurodegeneration after cerebral ischemia and may lead future research on Alzheimer’s disease or cerebral ischemia in new directions.

## 1. Cardiac Arrest

Cardiac arrest occurs as a result of a sudden stop of the heartbeat and its mechanical activity, which causes cessation of systemic circulation and blood flow in the brain, which triggers global brain ischemia [1,2]. A comprehensive analysis of studies conducted in Europe, China, and the USA showed that the average age of patients who suffered cardiac arrest was 60–66 years, and 58–81% of them were men [3,4,5]. However, rates of mortality, depression, anxiety, and cognitive impairment were higher in women than in men [6,7]. The incidence of cardiac arrest worldwide exceeds 3.7 million annually [8,9]. Among others, in Europe there are 275,000, in the USA 450,000, in Taiwan 10,000, of whom about 10–15% survive until discharge from hospital [10,11,12,13]. The annual incidence of in-hospital cardiac arrest is estimated at 35–55 per 100,000 in Europe and the USA and 28–43 per 100,000 in Asia [9,14,15,16]. In Europe, based on data from the EuReCa ONE and EuReCa TWO studies, the annual incidence of out-of-hospital cardiac arrest was estimated at 84 per 100,000 patients and 89 per 100,000 patients, respectively [17,18], with a moderate average survival rate of 8% [18,19]. In European countries, it is estimated that 56–62 people per 100,000 are treated by emergency services for out-of-hospital cardiac arrest each year [15,18]. In Latin America and Colombia, the epidemiological characteristics and treatment outcomes of patients with out-of-hospital cardiac arrest are similar to those described in the world literature [20]. Similar statistics apply to the United States, Australia, and New Zealand [21,22]. Generally, the return of spontaneous circulation after cardiac arrest ranges from 33.7 to 35.5% [5,23]. Male gender and advanced age have been shown to be independent factors for failure to return spontaneous circulation after cardiac arrest [3,4,5].

Patients who are successfully resuscitated suffer from cerebral hypoxia and ischemia, which is the main cause of side-effects and death after admission to the intensive care unit [24]. Brain damage after cardiac arrest is the leading cause of death in patients resuscitated after cardiac arrest and the leading cause of long-term disability in those who survive the acute phase [25,26]. Therefore, cardiopulmonary resuscitation is the most important therapeutic activity that restores life to patients after cardiac arrest [1,2]. Despite progress in the use of cardiopulmonary resuscitation and targeted brain temperature control by lowering the head temperature to 32–34 °C [12,27,28,29], only 10–15% of patients after cardiac arrest survive until discharge from the hospital [12,13]. Therefore, cardiac arrest and cardiopulmonary resuscitation constitute a serious challenge for emergency physicians around the world and their actions are crucial for patients’ survival. Two key prognostic indicators that are extremely important in post-cardiac arrest are the return of spontaneous systemic circulation and the patient’s neurological status after resuscitation. Unfortunately, cardiac arrest is characterized by a high incidence, limited return of spontaneous systemic circulation, poor neurological outcomes, and poor survival to discharge. Despite significant progress in pre-hospital and in-hospital care, sudden cardiac arrest is still characterized by high morbidity and mortality. Therefore, the impact of cardiac arrest on the quality of life is of great importance worldwide. We believe that the presented article on similar brain neurodegeneration following cardiac arrest and Alzheimer’s disease will both inform and stimulate further interest in the mechanisms of changes and the development of causal treatments for dementia after cerebral ischemia from cardiac arrest and in Alzheimer’s disease. Therefore, there is an urgent need to search for new molecular mechanisms and therapeutic strategies for patients after cardiac arrest. While the effect of cardiac arrest on the brain is obvious, the mechanisms underlying this process remain unclear [30]. The paper presents the neurodegeneration of the brain after cardiac arrest, with particular emphasis on the occurrence of cognitive disorders and dementia, and with attention to new proteomic and genomic data.

## 2. Brain Neurodegeneration after Cardiac Arrest

Brain neuropathology after cardiac arrest includes primary ischemic injury and secondary reperfusion injury, which occur sequentially, acutely during cardiac arrest and resuscitation, and chronically in the post-resuscitation stage [24]. Transient global brain ischemia resulting from cardiac arrest in humans and animals causes neurodegeneration of neurons in the hippocampus, brain cortex, amygdala, basal ganglia, thalamus, dorsal and lateral septum, olfactory tubercle, primary olfactory cortex, entorhinal cortex, and brainstem [2,13,30,31,32,33,34,35,36]. The hippocampus is one of the brain regions most susceptible to ischemia after cardiac arrest in humans [37,38,39] and in animals [40,41,42,43]. Transient global brain ischemia causes selective neurodegeneration in the CA1 area of the hippocampus in humans and animals within 2–7 days after reperfusion [30,42]. Two years after ischemia in animals, in addition to neuronal loss in ischemia-sensitive areas, various stages of neuronal pathology were also observed in other areas [42]. Acute and chronic neuronal changes have been demonstrated in brain areas unrelated to primary ischemic pathology, i.e., hippocampal areas CA2, CA3, and CA4 [42]. In the hippocampus, activation of glial cells precedes neuronal loss and continues for a long time after an ischemic event in animals [44,45] and humans [46]. In the brains of humans and animals after ischemia, changes in the white matter combined with the proliferation of glial cells have been documented [42,44,45]. Autopsy of brains after experimental ischemia with a survival time of up to 2 years and in humans after ischemia showed severe hippocampal atrophy [42,47,48]. These neuropathological alterations have been clearly associated with progressive cognitive decline in humans [48,49,50] as well as in animals [51,52].

Brain ischemia caused by cardiac arrest increases the permeability of the blood–brain barrier in humans [53] as well as animals to cellular and non-cellular blood elements, i.e., platelets [54] and amyloids [55]. In the case of post-ischemic blood–brain barrier leakage, two facts deserve attention: the first is related to the passage of amyloid into the brain, and the second is the penetration of, among others, platelets containing huge amounts of soluble amyloid, which causes additional neurotoxic damage to the brain parenchyma [42,56]. Additionally, the permeability of the blood–brain barrier was influenced by oxidative stress, neuroinflammation [45,46,57], and the *LRP1* and *RAGE* genes related to amyloid transport, described in detail later in the work. Soluble amyloid is delivered to the brain after cardiac arrest from the circulatory system [4,58] and additionally contributes to brain vessel vasoconstriction, amyloidosis, and cerebral amyloid angiopathy [49,50].

After brain ischemia, hippocampal atrophy was demonstrated with simultaneous neurodegenerative damage to the cortex of the temporal lobe [42,49,50]. Neuronal death after cerebral ischemia due to cardiac arrest may be dependent and independent of caspase. Moreover, neuronal death dependent on amyloid and modified tau protein has been demonstrated. Current preclinical and clinical evidence indicates that cardiac arrest causes chronic neuroinflammatory responses in the brain, which further exacerbate neurodegenerative changes [44,45,46]. Particularly extensive microglial activation and neurodegeneration in the CA1 area of the hippocampus in humans and animals are evident after cardiac arrest and lead to serious neurological sequelae [45,46,57].

## 3. Cognitive Deficits after Cardiac Arrest

The most sensitive brain regions to ischemic damage from cardiac arrest are the hippocampus, brain cortex, basal ganglia, thalamus, and amygdala [13,32,47]. These areas are closely related to cognitive domains (Figure 1) [13,51,52]. Cognitive functions depend on complex interactions between cortical and subcortical areas through various brain networks (Figure 1) [13]. Cognitive problems develop in 42–50% of patients who survive cardiac arrest up to several years after discharge from hospital [13,59]. Other studies have reported that in people who survived cardiac arrest, 50–100% of them experienced cognitive, mood, and functioning disorders [6,7,60]. Even among patients who returned to good neurological condition after discharge from the hospital, 29% experienced memory problems (short-term memory and spatial or contextual memory) and 43% experienced cognitive impairment [28,59].

The post-ischemic hippocampus is believed to be the main structure underlying episodic memory impairment, which is the earliest and most visible clinical symptom preceding the development of dementia (Figure 1) [42,48,49,50,51]. Ischemia causes damage to the temporal cortex, which is the target area of the main axon output network from the hippocampus, so these areas are structurally and functionally related to each other and are important for learning and memory processes [13,49,50]. Thus, existing evidence indicates that cognitive and functional impairment after cardiac arrest is one of the important areas of concern for physicians [61,62,63,64,65,66,67]. The primary goal of research in the past was to improve survival after cardiac arrest. Currently, in clinical conditions, there is a search for methods that can improve neurological outcomes. Therefore, it is not surprising that research has moved beyond prognostication in the acute period following successful cardiopulmonary resuscitation to identifying mechanisms of long-term brain damage. Another currently promising area of research is the implementation of therapies that can reduce neuronal damage resulting from cerebral hypoxia and ischemia. It should be added that it is unlikely that survivors could be considered fully recovered immediately after being discharged from the hospital. Typically, sooner or later during long-term follow-up, they develop cognitive deficits, mainly related to memory and executive functions, or mild to severe impairment, including memory loss, deterioration of psychomotor, executive, and visuospatial functions, and emotional problems, including anxiety and depression [28,68,69,70]. The above-mentioned disorders affect the vast majority of patients after cardiac arrest—estimated to be as many as 88% [71]. The most frequently stated cognitive deficit is memory loss in patients who regain consciousness after cardiac arrest. It should be noted that memory with delayed recall and recognition is usually most affected, while isolated memory loss is rare. It can be summarized that the typical pattern of impairment after cardiac arrest includes deficits in memory, fine motor skills, and in executive functions [72]. According to the latest recommendations, before leaving the hospital, patients should be thoroughly assessed for short- and long-term memory deficits, cognitive and executive dysfunction, depression, and possible progression to dementia [62]. Overall, the prognosis for recovery of cognitive function to pre-arrest levels is uncertain. Despite uncertain prognoses, long-term rehabilitation has been shown to improve overall performance across a wide range of activities and should be pursued [73]. In most cases, patients are unable to return to daily activities within the first year after injury, and deficiencies are detected up to 8 years after cardiac arrest [74,75]. A study from Sweden, with an extremely long-term follow-up of 17 years after cardiac arrest, found a trend towards lower scores on cognitive tests and lower self-reported quality of life [71]. The authors concluded that survivors of cardiac arrest may experience permanent cognitive impairment, progressing to Alzheimer’s disease-type dementia [71,76]. It is suggested that the above-mentioned population is at increased risk of dementia due to the hypoxia occurring during cardiac arrest. It has been noted that the progression of cognitive decline following cardiac arrest in humans is characterized by greater memory impairment than executive dysfunction, making it similar to the prodromal form of Alzheimer’s disease dementia [76].

Cardiac arrest in humans and animals has been shown to significantly increase the risk of mild cognitive impairment, vascular cognitive impairment, and dementia, and dementia of Alzheimer’s disease type [46,51,52,77,78]. More than a quarter of patients experienced cognitive impairment four years after cardiac arrest [46,79]. Cohort studies have shown a high prevalence (54.4%) of long-term cognitive deficits and functional limitations in cardiac arrest survivors [46,80], even in those with apparently favorable neurological outcomes [46,81].

## 4. Alzheimer’s Disease Related Genes after Cardiac Arrest

It has been shown that in the CA1 and CA3 regions of the hippocampus, as well as in the temporal cortex, genes associated with Alzheimer’s disease, such as these encoding *α-secretase*, *amyloid precursor protein*, *β-secretase*, *presenilin 1* and *2*, and *tau protein* are dysregulated as a result of ischemia caused by cardiac arrest (Table 1) [82]. Changes in gene expression in the CA1 area of the hippocampus include all genes examined at all time observations after cardiac arrest (Table 1) [82]. However, changes in gene expression in the CA3 area are less severe, do not occur at all times after ischemia, and are delayed in time in relation to the CA1 region (Table 1) [82]. Specifically, changes in gene expression in the temporal cortex occurred immediately after cardiac arrest but did not occur at all times after ischemia and did not affect all genes (Table 1) [82]. The results indicate that a brain ischemic episode caused by cardiac arrest is a trigger of amyloidogenic processes in brain parenchyma [42,47,83,84,85]. After cardiac arrest, amyloidogenic processing of amyloid protein precursor was observed with varying intensity depending on the structure. The consequences are additional changes in ischemic neurons depending on the toxic properties of amyloid. This creates conditions for the accumulation of amyloid plaques in the brain as in Alzheimer’s disease.

## 5. Amyloid and Tau Protein in Humans after Cardiac Arrest

Studies of human brains after cardiac arrest have shown intense accumulation of amyloid in the brain parenchyma [86]. The presence of both diffuse and senile amyloid plaques has been demonstrated in areas of the brain susceptible to ischemia, in the border zones of arteries, and in the brain cortex [86]. Intense amyloid accumulation was noted in the middle layers of the brain cortex, which are sensitive to ischemia [86]. In the brains of patients with survival up to 1 month after cardiac arrest, strong amyloid staining was found in neurons and perivascular areas [86]. The neurons of the brain cortex and hippocampus stained most intensively for amyloid. Ependymal and epithelial cells also stained for amyloid. The brain’s gray and white matter vessels were surrounded by numerous amyloid deposits that looked like cuffs. In some brains, the walls of meningeal and cortical vessels were stained for amyloid. The accumulation of amyloid in the perivascular spaces of the blood–brain barrier vessels suggests that the amyloid originated from the blood. Evidence supporting this hypothesis comes from clinical studies indicating the presence of significantly elevated blood amyloid levels in patients after cardiac arrest [4,58]. The level of amyloid increase correlated negatively with clinical outcome after cerebral ischemia, and this, in turn, likely indicates the degree of brain damage [4,58]. These data confirm that brain ischemia resulting from cardiac arrest may play a key role in the amyloidogenic metabolism of the amyloid precursor protein.

Tau protein was detected in the blood of patients after complete brain ischemia due to cardiac arrest with two peaks on the second and fourth day, which probably indicates the degree of neuronal injury [4,87]. The detected two-phase changes in the level of tau protein in the blood correlate with two types of neuronal death after ischemia, as a result of necrosis and delayed neuronal death [88]. The presented studies show that the concentration of tau protein in the serum may be a biomarker for assessing brain damage after cardiac arrest [87,88,89,90].

## 6. Amyloid and Tau Protein in Animals after Cardiac Arrest

After experimental brain ischemia resulting from cardiac arrest with survival of 1 year, various fragments of amyloid precursor protein were found in the extra- and intracellular spaces [42,47,83]. In rats that survived up to 0.5 years after brain ischemia, N- and C-terminal deposits of amyloid precursor protein and amyloid were visible in the extracellular space of the hippocampus, brain cortex, white matter, and around the lateral ventricles [42,83]. Various parts of the amyloid precursor protein have also been found to accumulate in neuronal cells, neuroglia, endothelium, pericytes, and ependymal cells [42,47,85]. In particular, astrocytes around the microvessels showed intense staining of very long, thin processes that adhered to or surrounded the capillaries. However, after 0.5 years of post-ischemic survival, only staining of the C-terminal of the amyloid precursor protein and amyloid was noted [42]. Amyloid accumulation in response to reversible brain ischemia was not a transient phenomenon, as diffuse amyloid deposits converted to amyloid plaques in animals with 1-year survival [84].

Evidence shows that after ischemia, neuronal cells are dominated by amyloid accumulation and hyperphosphorylated tau protein, which accompany apoptosis [42,91,92,93]. It is also evident that paired helical filaments [94], neurofibrillary tangles-like [95], and neurofibrillary tangles were observed in neurons after ischemia [96,97,98]. Data indicate that brain ischemia additionally activates neuronal death in an amyloid- and tau-protein-dependent manner [96,97,98].

## 7. Transport Genes of Amyloid and Tau Protein in the Brain after Cardiac Arrest

The expression of the *receptor for the advanced glycation end products* (*RAGE*) gene increases significantly in CA3 area during 7–30 days after ischemia, and at 12–24 months its expression gradually and significantly decreases (Table 2) [99]. Regarding the expression of the *low-density lipoprotein receptor-related protein 1* (*LRP1*) gene, the opposite is true (Table 2). In the early phase post-ischemia (2–7 days), a significant decrease in its expression was observed [99]. Then, 12–24 months after cardiac arrest, its expression gradually increased and was statistically significant [99]. In other words, in the early phase after ischemia, 7–30 days, the expression of the gene *RAGE* transporting amyloid and tau protein to the brain increased with increase in the neurotoxic activity of amyloid and tau protein. Conversely, in the late phase after ischemia, i.e., 12–24 months, the activity of the amyloid clearing gene *LRP1* increased spontaneously in the CA3 area, reducing and/or preventing further neuronal injury or facilitating healing (Table 2).

In the brains of patients after ischemia caused by cardiac arrest, immunostaining for the receptor for advanced glycation end products was localized in the epithelial cells of the choroid plexus as well as in the cells of the lining of the brain ventricles [100]. The above cells form the cerebrospinal fluid–brain barrier and the blood–cerebrospinal fluid barrier. In these cases, amyloid staining was observed in the blood vessels of the choroid plexus and the basement membrane of the choroid plexus epithelium [100]. These data indicate that the choroid plexus epithelium and cells lining the ventricles are equipped with the advanced glycation end-product receptor and play a significant role in the transport and accumulation of amyloid in the brain parenchyma.

## 8. Genes Involved in Neuronal Death in Cardiac Arrest and Alzheimer’s Disease

*Autophagy* (*BECN1*) gene expression in the CA1 area of the hippocampus after cardiac arrest with survival at 2, 7, and 30 days was within control limits (Table 3) [101]. However, *mitophagy* (*BNIP3*) gene expression 2 days after ischemia in CA1 was above the control value, and on days 7 and 30 after cardiac arrest, it was within the control range (Table 3). As for the expression of the *apoptosis* (*caspase 3*) gene in the CA1 area, its overexpression was found after 2 and 7 days of survival, while on the 30th day after cardiac arrest, the expression was below the control value (Table 3) [101].

*BECN1* gene expression in CA3 increased significantly on day 30 after cardiac arrest (Table 3) [102]. However, *BNIP3* gene expression after cardiac arrest was below control values for 2–30 days of observation (Table 3) [102]. Studies of the *caspase 3* gene showed increased activity on days 7 and 30 after cardiac arrest (Table 3) [102].

*BECN1* gene expression in the temporal cortex 2 days after cardiac arrest was above the control value, while after 7 and 30 days it oscillated around control values (Table 3) [103]. However, 2 days after cardiac arrest, the expression of the *BNIP3* gene was below the control value, and after 7 and 30 days its expression increased significantly above the control value (Table 3) [103]. As for the expression of the *caspase 3* gene in this structure, 2 days after cardiac arrest, it was below the control value, and on days 7 and 30, its expression was above the control value (Table 3) [103].

## 9. Discussion

Harvested evidence suggests that there is a remarkable parallelism between the neuropathogenesis of Alzheimer’s disease and brain ischemia in animals and humans resulting from cardiac arrest [42,104,105,106,107,108,109,110,111]. Both the Alzheimer’s disease and post-cardiac arrest brain are characterized by neuronal death, blood–brain barrier permeability, neuroinflammation, amyloid plaques, neurofibrillary tangles, cerebral amyloid angiopathy, and hippocampal and brain atrophy [42,44,45,49,50,84,85,93,96,97]. Collectively, these types of evidence point to common proteomic and genomic mechanisms of cerebral ischemia caused by cardiac arrest and Alzheimer’s disease [42,44,45,82,84,85,96,97,98].

After cardiac arrest, neuronal death is observed in the hippocampus, cerebral cortex, basal ganglia, thalamus, and amygdala, structures responsible for memory, as well as general brain atrophy and, ultimately, progressive dementia of Alzheimer’s disease phenotype (Figure 1) [51,52,59,77]. One proposal explaining the above changes suggests that long non-coding RNA increases the expression of the adapter protein ShcA in the hippocampus, which causes cognitive impairment after cardiac arrest as a result of neuroinflammation and apoptosis of pyramidal neurons [112,113,114]. Another hypothesis links hypoxia to molecular abnormalities of the cerebral cortex resembling Alzheimer’s disease, which may facilitate the development of this dementia phenotype. It has been shown that transient hypoxia is associated with hyperphosphorylation of cytoskeletal proteins and abnormalities in the functioning of neuronal mitochondria, which may lead to Alzheimer’s disease-like changes and may also cause an increase in the level of amyloid, which is an important element in the pathology of Alzheimer’s disease [115]. Yet another possible pathway leading to the pathogenesis of a post-ischemic Alzheimer’s disease phenotype is dysregulation of the expression of genes that metabolize the amyloid precursor protein to amyloid and modification of the tau protein, observed after global cerebral ischemia in an experimental model of cardiac arrest [116,117,118,119,120]. Brain ischemia is the second naturally occurring brain pathology after Alzheimer’s disease, causing the death of pyramidal neurons in the hippocampus [42]. An increase in amyloid levels in the brain and blood indicates activation of the amyloidogenic pathway in response to acute cerebral ischemia caused by cardiac arrest. The above observations allow us to suggest that brain ischemia may be the cause of Alzheimer’s disease [42,45,93,104,105,107,108,109,110,111].

Currently, a small group of scientists specifically proposes the involvement of ischemia in the development of Alzheimer’s disease at three levels: cellular, brain, and patient [42,104,105,106,107,108,109,110,111]. They discuss the involvement of ischemia in Alzheimer’s disease as dysregulation of neurovascular coupling, reduced cerebral blood flow, blood–brain barrier permeability, and reduced amyloid clearance from the brain. Finally, they present a two-step etiology of Alzheimer’s disease that involves a two-step positive feedback loop in which cerebral microvascular damage(s) precedes the production and accumulation of amyloid and tau protein abnormalities in the brain [42,104,105,106,107,108,109,110,111].

## 10. Conclusions

In previous studies, the main goal was to increase survival after cardiac arrest. Unfortunately, the current high incidence of cognitive impairment observed in cardiac arrest survivors remains a significant challenge for both the research and clinical communities. It seems certain that future research should focus on the prevention of neurological damage and effective therapeutic approaches that could improve neurological outcomes after cardiac arrest. Future research efforts should focus on finding pharmacological and nonpharmacological interventions to improve cognitive function, mental health, and quality of life in cardiac arrest survivors. The conclusions drawn from the study of Alzheimer’s disease-related proteins after cardiac arrest and their genes in the hippocampus and temporal cortex, which contribute to neuronal death, amyloid production, and the formation of neurofibrillary tangles, are important for the development of therapeutic targets also in the treatment of Alzheimer’s disease.

Finally, the results clearly indicate that changes in the expression of the presented genes play an important role in acute and secondary brain damage and the development of post-ischemic brain neurodegeneration with the Alzheimer’s disease phonotype. The data indicate that the above genes may be a potential therapeutic target for brain therapy after ischemia due to cardiac arrest. Overall, the studies show that the genes studied represent attractive targets for the development of new therapies to minimize ischemic brain injury and neurological dysfunction. Additionally, *amyloid*-related genes expression and *tau protein* gene modification after cerebral ischemia due to cardiac arrest are useful in identifying ischemic mechanisms associated with Alzheimer’s disease. Cardiac arrest illustrates the progressive, time- and area-specific development of neuropathology in the brain with the expression of genes responsible for the processing of *amyloid protein precursor* and the occurrence of *tau protein* and symptoms of dementia, such as those occurring in patients with Alzheimer’s disease. By carefully examining the common genetic processes involved in these two diseases, these data may help unravel phenomena associated with the development of Alzheimer’s disease and neurodegeneration after cerebral ischemia and may lead future research on Alzheimer’s disease or cerebral ischemia in new directions.

## Figures and Tables

**Figure 1 ijms-25-01291-f001:**
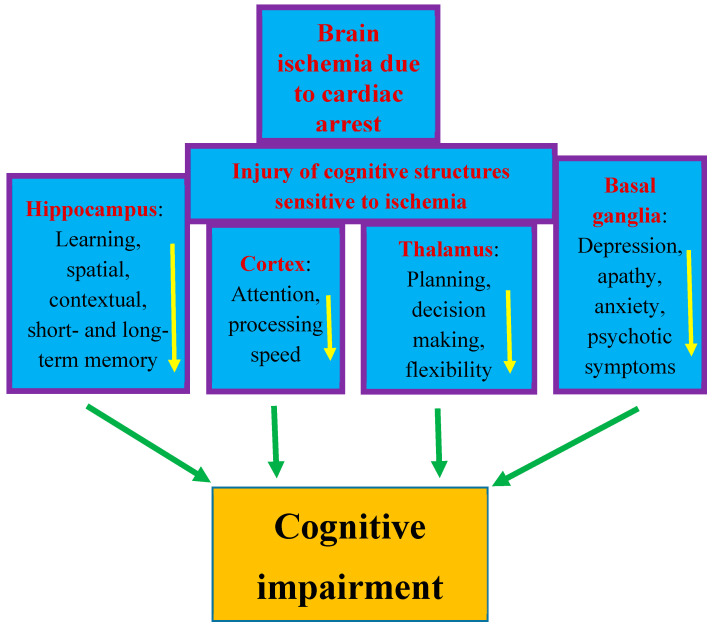
Structures susceptible to ischemic brain injury due to cardiac arrest and responsible for cognition. The figure shows damage to structures affecting memory, behavior, and cognitive impairment after cardiac arrest in an identical manner to that seen in Alzheimer’s disease. Arrows indicate decreased activity.

**Table 1 ijms-25-01291-t001:** Changes in the expression of the Alzheimer’s disease-linked genes in the hippocampus and temporal cortex at various times after ischemic brain injury due to cardiac arrest.

Genes Survival	*APP*	*ADAM10*	*BACE1*	*PSEN1*	*PSEN2*	*MAPT*
CA1 subfield of hippocampus						
2 days	↑	N.A.	↑↑	↑	↑↑	↑↑
7 days	↑	N.A.	↑	↑	↑	↔
30 days	↑	N.A.	↓	↓	↓	↔
CA3 subfield of hippocampus						
2 days	↔	↓	↓	↑	↔	↔
7 days	↑	↓	↓	↑	↓	↑
30 days	↔	↓	↑	↔	↑	↑
Medial temporal cortex						
2 days	↓	N.A.	↑↑	↔	↑↑	N.A.
7 days	↑	N.A.	↔	↔	↔	N.A.
30 days	↑	N.A.	↔	↔	↔	N.A.

Expression: ↑ increase; ↓ decrease; ↔ oscillation around control values; N.A., not available. Genes: *APP*, *amyloid protein precursor*; *ADAM10*, *α-secretase*; *BACE1, β-secretase*; *PSEN1*, *presenilin 1*; *PSEN2*, *presenilin 2*; *MAPT*, *tau protein*.

**Table 2 ijms-25-01291-t002:** Changes in the expression of amyloid and tau protein transporting genes in the hippocampal CA3 area at various times after ischemic brain injury due to cardiac arrest.

Survival Genes	2 Days	7 Days	30 Days	12 Months	18 Months	24 Months
*LRP1*	↓	↓	↔	↑	↑↑	↑
*RAGE*	↓	↑	↑	↓↓	↓	↓

Expression: ↑ increase; ↓ decrease; ↔ oscillation around control values. Genes: LRP1-low-density lipoprotein receptor-related protein 1; RAGE-receptor for advanced glycation end products.

**Table 3 ijms-25-01291-t003:** Changes in the expression of *autophagy*, *mitophagy*, and *apoptosis* genes linked to Alzheimer’s disease in the hippocampus and temporal cortex at various times after ischemic brain injury due to cardiac arrest.

Genes Survival	*BECN1*	*BNIP3*	*CASP3*
CA1 subfield of hippocampus			
2 days	↔	↑	↑↑↑
7 days	↔	↔	↑
30 days	↔	↔	↓
CA3 subfield of hippocampus			
2 days	↔	↓	↓
7 days	↓	↓	↑
30 days	↑	↓	↑
Medial temporal cortex			
2 days	↑	↓↓	↓
7 days	↔	↑↑↑	↑
30 days	↔	↑	↑

Expression: ↑ increase; ↓ decrease; ↔ oscillation around control values. Genes: *BECN1*-*autophagy*; *BINP3*-*mitophagy*; *CASP3-apoptosis.*

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
