# Peer review of "Ischemia-Reperfusion Programming of Alzheimer’s Disease-Related Genes—A New Perspective on Brain Neurodegeneration after Cardiac Arrest"

_ijms, 2024, doi:10.3390/ijms25021291_

Round 1

Reviewer 1 Report

Comments and Suggestions for Authors

1. Scope and Relevance: The article effectively addresses the pertinent issue of cerebral ischemia following cardiac arrest and its potential link to Alzheimer's disease (AD) pathogenesis. It offers comprehensive insights into the molecular and genetic alterations in the brain post-cardiac arrest, which is a novel and significant area of research with implications for understanding and treating both cardiac arrest outcomes and Alzheimer's disease.

2. Methodological Strengths:

  • The study is well-structured, systematically exploring various facets such as gene expression changes in ischemic brain regions, the accumulation of amyloid and tau proteins, and the implications of these changes for cognitive function and AD pathology.
  • The integration of data from both human and animal models provides a robust framework for understanding the ischemia-induced neurodegenerative processes.

3. Key Findings:

  • The evidence of parallel neuropathological features between AD and brain ischemia post-cardiac arrest is compelling and thought-provoking.
  • The detailed analysis of gene expression related to amyloid and tau proteins post-ischemia offers valuable insights into the molecular underpinnings of neurodegeneration.

4. Areas for Improvement and Clarification:

  • While the article presents a comprehensive overview, it would benefit from a more in-depth discussion of the mechanistic links between ischemia-induced changes and Alzheimer's disease progression. For instance, exploring the potential pathways through which ischemia may exacerbate or trigger AD-like neurodegeneration could provide a more holistic understanding.
  • The paper could also delve deeper into the implications of these findings for clinical practice, particularly in terms of potential therapeutic targets and preventive strategies for cardiac arrest survivors at risk of developing Alzheimer’s disease.

5. Overall Impression: This article is a valuable contribution to the field, bridging the gap between cardiac arrest-induced brain ischemia and Alzheimer's disease. It opens new avenues for research into therapeutic interventions that could ameliorate the long-term neurological outcomes of cardiac arrest survivors. Future studies, as suggested by the authors, should focus on identifying effective interventions to prevent or mitigate the neurological damage post-cardiac arrest, which could have significant implications for patient care and rehabilitation.

Author Response

All changes are in red.

Reviewer 1.

  1. Scope and Relevance: The article effectively addresses the pertinent issue of cerebral ischemia following cardiac arrest and its potential link to Alzheimer's disease (AD) pathogenesis. It offers comprehensive insights into the molecular and genetic alterations in the brain post-cardiac arrest, which is a novel and significant area of research with implications for understanding and treating both cardiac arrest outcomes and Alzheimer's disease.

Thanks.

  1. Methodological Strengths:

The study is well-structured, systematically exploring various facets such as gene expression changes in ischemic brain regions, the accumulation of amyloid and tau proteins, and the implications of these changes for cognitive function and AD pathology.

The integration of data from both human and animal models provides a robust framework for understanding the ischemia-induced neurodegenerative processes.

Thanks.

  1. Key Findings:

The evidence of parallel neuropathological features between AD and brain ischemia post-cardiac arrest is compelling and thought-provoking.

The detailed analysis of gene expression related to amyloid and tau proteins post-ischemia offers valuable insights into the molecular underpinnings of neurodegeneration.

Thanks.

  1. Areas for Improvement and Clarification:

While the article presents a comprehensive overview, it would benefit from a more in-depth discussion of the mechanistic links between ischemia-induced changes and Alzheimer's disease progression. For instance, exploring the potential pathways through which ischemia may exacerbate or trigger AD-like neurodegeneration could provide a more holistic understanding.

Done.

The paper could also delve deeper into the implications of these findings for clinical practice, particularly in terms of potential therapeutic targets and preventive strategies for cardiac arrest survivors at risk of developing Alzheimer’s disease.

Done. In conclusions we added some information.

  1. Overall Impression: This article is a valuable contribution to the field, bridging the gap between cardiac arrest-induced brain ischemia and Alzheimer's disease. It opens new avenues for research into therapeutic interventions that could ameliorate the long-term neurological outcomes of cardiac arrest survivors. Future studies, as suggested by the authors, should focus on identifying effective interventions to prevent or mitigate the neurological damage post-cardiac arrest, which could have significant implications for patient care and rehabilitation.

Thanks.

Reviewer 2 Report

Comments and Suggestions for Authors

The manuscript entitled Ischemia-reperfusion programming of Alzheimer's disease-related genes - a new perspective on brain neurodegeneration after cardiac arrest, brings a new inside in the mechanisms associated with Alzheimer's disease and amyloid plaque accumulation into the brain.

Observations

Abstract

Please present more details about the subject in your manuscript. The abstract is to short and is nor attractive for the readers. If it is an review you need to mention what kind of review is your paper. 

Introduction 

Last sentence of the first paragraph - indicate the reference, please.

Chapter 2

Please indicate the mechanism of BBB permeabilisation, what kind of molecules contribute to this, the structure of BBB and the dynamic process of amyloid accumulation in the brain tissue. The contribution of oxidative stress and inflammation is important in this process, therefore is important to describe details about these processes. Describe also the ischemia /reperfusion process after cardiac arrest because this is the main mechanism of your manuscript, which is leading to BBB permeabilization and, consequently, to amyloid deposition. It is also important to describe the amyloid metabolism.

In the paragraph where you describe the cerebral atrophy you should mention the neuronal death mechanisms, because cerebral atrophy consequently to neuronal death. There are several neuronal death associated with neuronal ischemia. Please describe them. I suggest a table for this idea. 

Fig 1 - the words are not quite visible because of improper contrast between the letters and background. Please adjust it.  Please explain more details about processes in this figure, in the text below.

Chapter 3

Indicate eventually which memory is affected.

Conclusions

Please do not indicate the references in the Conclusions chapter. Move that sentences in a proper chapter above. The Conclusions has to contain only your main idea of manuscript, why the mechanisms you described are important for AD diagnosis, and what are the treatment perspectives connected with these mechanisms. 

Author Response

All changes are in red.

Reviewer 2.

The manuscript entitled Ischemia-reperfusion programming of Alzheimer's disease-related genes - a new perspective on brain neurodegeneration after cardiac arrest, brings a new inside in the mechanisms associated with Alzheimer's disease and amyloid plaque accumulation into the brain.

Thanks.

Observations

Abstract

Please present more details about the subject in your manuscript. The abstract is to short and is nor attractive for the readers.

Done.

 If it is an review you need to mention what kind of review is your paper. 

It is “Perspective”.

Introduction 

Last sentence of the first paragraph - indicate the reference, please.

Done.

Chapter 2

Please indicate the mechanism of BBB permeabilisation, what kind of molecules contribute to this, the structure of BBB and the dynamic process of amyloid accumulation in the brain tissue. The contribution of oxidative stress and inflammation is important in this process, therefore is important to describe details about these processes. Describe also the ischemia /reperfusion process after cardiac arrest because this is the main mechanism of your manuscript, which is leading to BBB permeabilization and, consequently, to amyloid deposition. It is also important to describe the amyloid metabolism.

Data on amyloid metabolism are scattered throughout the manuscript depending on the topic covered. This part contains information about its passage and the passage of platelets rich in amyloid and amyloid precursor protein through the open blood-brain barrier. In addition, impact of oxidative stress and neuroinflammation has been added. Here it is necessary to apologize the reviewer and explain that the aim of the study was not the mechanisms of opening the blood-brain barrier, but the fact of opening and what pathological substances pass through the barrier. Please note that the end of this section contains information about neuroinflammation after cardiac arrest from the beginning.

In the paragraph where you describe the cerebral atrophy you should mention the neuronal death mechanisms, because cerebral atrophy consequently to neuronal death. There are several neuronal death associated with neuronal ischemia. Please describe them. I suggest a table for this idea. 

We added some information about neuronal death which strictly connected with cardiac arrest.

Fig 1 - the words are not quite visible because of improper contrast between the letters and background. Please adjust it.  Please explain more details about processes in this figure, in the text below.

Done.

Chapter 3

Indicate eventually which memory is affected.

Done.

Conclusions

Please do not indicate the references in the Conclusions chapter. Move that sentences in a proper chapter above. The Conclusions has to contain only your main idea of manuscript, why the mechanisms you described are important for AD diagnosis, and what are the treatment perspectives connected with these mechanisms. 

Done.